# Streaming Kernel PCA Algorithm With Small Space

Yichuan Deng[1], Jiangxuan Long[2], Zhao Song[3], Zifan Wang[4], Han Zhang[5]

[1]University of Science and Technology of China, [2]South China University of Technology, [3]The Simons Institute for the Theory of Computing at the University of California, Berkeley, [4]Stonybrook University, [5]University of Washington.

`ethandeng02@gmail.com, lungchianghsuan@gmail.com, magic.linuxkde@gmail.com,`
`Zifan.wang@stonybrook.edu, micohan@cs.washington.edu`

Principal Component Analysis (PCA) is a widely used technique in machine learning, data analysis, and signal processing. With the increase in the size and complexity of datasets, it has become essential to develop low-space usage algorithms for PCA. Streaming PCA has gained significant attention in recent years, as it can handle large datasets efficiently. The kernel method, commonly used in learning algorithms such as Support Vector Machines (SVMs), has also been applied in PCA algorithms.

We propose a streaming algorithm for Kernel PCA problems based on the traditional scheme by Oja. Our algorithm addresses the challenge of reducing the memory usage of PCA while maintaining its accuracy. We analyze the performance of our algorithm by studying the conditions under which it succeeds. Specifically, we show that when the spectral ratio $R := \lambda_1/\lambda_2$ of the target covariance matrix is $\Omega(\log n \cdot \log d)$, the streaming PCA can be solved with linear space cost. However, the standard PCA algorithm usually requires quadratic space due to matrix vector multiplication.

Our proposed algorithm has several advantages over existing methods. First, it is a streaming algorithm that can handle large datasets efficiently. Second, it employs the kernel method, which allows it to capture complex nonlinear relationships among data points. Third, it has a low-space usage, making it suitable for limited memory applications.

## 1. Introduction

Principal Component Analysis (PCA) is a technique used to reduce the dimension of data. PCA has been widely applied in various domains, including web-related applications [1], computer vision [2], and recommendation systems [3]. It is a linear method that uses orthogonal transformations to convert a set of correlated variables into a set of less correlated variables called *principal components*. In the simplest case, we care about the first principal component.

Kernel principal component analysis (kernel PCA) is an extension (also a generalization) of PCA, combined with the kernel methods. Kernel PCA has many applications, such as distance-based algorithm [4], computing principal components in high-dimensional feature spaces [5], face recognition [6, 7], spectral embedding [8], novelty detection [9], de-noising in feature spaces [10], and fault detection and identification of nonlinear processes [11].

In the simplest setting of PCA, given a dataset $X = \{x_1, x_2, \ldots, x_N\} \subseteq \mathbb{R}^d$, thus the *covariance matrix* of the dataset is $C := \frac{1}{N} \sum_{i \in [N]} x_i x_i^\top$. The goal is to find the eigenvector $v^* \in \mathbb{R}^d$ corresponding to the largest eigenvalue $\lambda$ of $C$.

To understand the motivation of kernel PCA [10, 12, 13], particularly for clustering, observe that, while $N$ points cannot, in general, be linearly separated in $d < N$ dimensions, they can almost always be linearly separated in $d \geq N$ dimensions. That is, given $N$ points, $x_i$, if we map them to an $N$-dimensional space with $\phi(x_i)$, where $\phi : \mathbb{R}^d \to \mathbb{R}^N$, it is easy to construct a hyperplane that

Second Conference on Parsimony and Learning (CPAL 2025).

divides the points into arbitrary clusters. So Kernel PCA is a widely-used tool to extract *nonlinear* features while traditional (linear) PCA can only detect *linear* features.

Since the dimension might be very high in the kernel space (implied by the kernel function $\phi$), computing the exact products in that space will be too expensive. Thus it is natural and reasonable to use *Mercer kernels* [14–16], a function $k(x, y) : \mathbb{R}^d \times \mathbb{R}^d \to \mathbb{R}_+$ such that, for an input dataset $X = \{x_i\}_{i \in [N]} \subseteq \mathbb{R}^d$, it produces a positive matrix $K \in \mathbb{R}^{n \times n}$, where each entry of $K$ is given by $K_{i,j} := k(x_i, x_j)$. By defining $k(x, y) := \phi(x)^\top \phi(y)$, one can use $k$ to map the data points to the kernel space without computing the inner product explicitly. Note that, each column $K_i$ of the matrix $K$ is the product in the kernel space from one point $x_i$ to all the $N$ points in $X$.

Since we don't work in the feature space explicitly (which might be very expensive due to the dimension), the principal components that have been found are for the projected data. For a data point $x$, its projection onto the $k$-th principal component $v_k$ is $v_k^\top \phi(x)$ instead of the original $v_k^\top x$ in the linear PCA.

In traditional PCA problem [17–20], one needs to have access to all the data points $\{x_i\}_{i \in [n]}$. Thus the space needed might be very high to store in memory. Streaming PCA is a method for performing PCA on data too large to fit into memory. The traditional PCA algorithm requires that all data is loaded into memory at once, making it infeasible for very large datasets. Streaming PCA, on the other hand, allows data to be processed in smaller chunks, reducing memory requirements and making it possible to analyze very large datasets.

In the streaming setting, we are asked to maintain a data structure such that, it receives the data points coming in the streaming way, and it can output the estimated principal component at the end of the streaming. Formally, the data structure receives a stream of $x_i$'s. Then with some maintaining operation, it can output a vector $u$ such that $u \approx x^*$, where $x^*$ is the top principal component of the dataset.

With the motivation of kernel PCA algorithm, combining the natural expectation for an algorithm to run fast/use low space, we ask the question

*Can we solve the kernel PCA in a small space?*

In this work, we present a positive answer to this problem.

## 1.1. Related Work

**Streaming Algorithms.** Over the past decades, a massive number of streaming algorithms have been designed, since there is a concern that under some circumstances, the data is too large to store in a single machine. Some streaming algorithms are mainly designed for graph problems [21], for instances, shortest path and diameter [22, 23], maximal independent set [24, 25], maximum matching and minimum vertex cover [22, 26, 27], spectral sparsification [28, 29], max-cut [30], kernel method and sketching technique [31–35]. Beyond graphs, streaming algorithms also provide insights into other fields, like the multi-armed bandit problem [36]. Since many problems are provably to be intractable with sublinear space of $n$, where we use $n$ to denote the number of nodes in the graph, a line of work [22, 37] has been focused on *semi-streaming* model. In this setting, the streaming algorithm is allowed to use $O(n \operatorname{poly} \log n)$ space.

Recently, attention has been focused on the streaming models under the setting of *multi-pass*, where under this setting, the models are allowed to look at the streaming updates more than once. The reason is that it can reduce the space needed effectively to let the models take more than one pass of the updates. For instances, an $O(\log \log n)$-pass algorithm for maximal independent set [24, 25, 38], and $O(1)$-pass algorithm for approximate matching [26, 27, 39, 40].

**Principal Component Analysis.** There has been a lot of research looking at Principal Component Analysis from a statistical point of view, where the performance of different algorithms is studied under specific conditions. This includes using generative models of the data [17], and making

assumptions about the eigenvalue spacing [18] and covariance matrix spectrum [19, 20]. While these studies do offer guarantees for a finite amount of data, they are not practical for real-world applications, as they are either limited to only working with a complete dataset or require a lot of computational resources. An efficient, incremental algorithm is needed for practical use.

Talking about incremental algorithms, the work of Warmuth and Kuzmin [41] provides an analysis of the worst-case streaming PCA. Previous general-purpose incremental PCA algorithms have not been analyzed for their performance with a finite amount of samples. [42]. Recently, there have been efforts to address the issue of lacking finite-sample analysis by relaxing the nonconvex nature of the problem. [43] or making generative assumptions [44].

As it is an attractive topic (it is natural to ask to extract principal components from a dataset coming in a streaming fashion), attention has been focused on streaming PCA for years. There are two traditional algorithms for streaming PCA, one is Oja's algorithm [45] and the other is a classical scheme provided by Krasulina [46]. The work of Balsubramani, Dasgupta and Freund [47] analyzes the rate of convergence of the Krasulina and Oja algorithms. The work by Hardt and Price [48] provided a robust convergence analysis of the well-known power method for computing the dominant singular vectors of a matrix that we call the noisy power method. Later work of Allen-Zhu and Li [49] provides global convergence for Oja's algorithm with $k > 1$ top principal components, and provides a variant of Oja's algorithm which runs faster. Another line of works [50, 51] shows that Oja's algorithm achieves performance nearly matching that of an optimal offline algorithm even for updates not only rank-1. There are also works focused on the problem of uncertainty quantification for the estimation error of the leading eigenvector from Oja's algorithm [52]. A very recent work [53] gives the correctness guarantee that under some specific conditions for the spectral ratio, Oja's algorithm can be used to solve the streaming PCA under a traditional setting.

## 1.2. Our Result

Here in this section, we present our main result, which is a streaming algorithm for kernel PCA.

---

**Algorithm 1** Our Streaming Kernel PCA Algorithm

---

1: **procedure** KERNELPCA$(n, d, m, \phi)$                                                     $\triangleright$ Theorem 1.1
2:      $v_0 \sim \mathcal{N}(0, I_m)$                              $\triangleright$ To store $v_0$ we only need $O(m)$ space.
3:      **for** $i = 1 \to n$ **do**
4:          Receive $x_i$
5:             $\triangleright$ To store $x_i$ we need $O(d)$ space, once we move to iteration $i + 1$, we can drop the $x_i$. Thus overall, we only need $O(d)$ space
6:          $v_i \leftarrow v_{i-1} + \eta \cdot \langle \phi(x_i), v_{i-1} \rangle \cdot \phi(x_i)$
7:             $\triangleright$ Over the entire algorithm we only need $O(m)$ space to store $v_i$. Once we move to $i + 1$, we don't need $v_{i-1}$ anymore
8:      **end for**
9:      $u \leftarrow v_n$
10:     **return** $u$
11: **end procedure**

---

**Theorem 1.1** (Informal version of Theorem 3.2). *Let $\phi : \mathbb{R}^d \to \mathbb{R}^m$. Let $\Sigma = \frac{1}{n} \sum_{i=1}^{n} \phi(x_i)\phi(x_i)^\top \in \mathbb{R}^{m \times m}$. We define $R := \lambda_1(\Sigma)/\lambda_2(\Sigma)$ where $\lambda_1(\Sigma)$ is the largest eigenvalue of $\Sigma$ and $\lambda_2(\Sigma)$ is the second largest eigenvalue of $\Sigma$. Let $x^*$ denote the top eigenvector of $\Sigma$. Let $C > 10^3$ denote a sufficiently large constant. If $R \geq C \cdot (\log n) \cdot (\log d)$, there is a streaming algorithm (Algorithm 1) that only uses $O(d + m)$ spaces and receives $x_1, x_2, \cdots, x_n$ in the online/streaming fashion, and outputs a unit vector $u$ such that*

$$1 - \langle x^*, u \rangle^2 \leq (\log d)/R$$

*holds with probability at least $1 - \exp(-\Omega(\log d))$.*

By combining the kernel method and the streaming PCA technique, Algorithm 1 provides a way to solve kernel PCA with linear cost of space when the spectral ratio $R = \Omega(\log n \cdot \log d)$, as we show in Theorem 1.1.

**Roadmap.** In Section 2, we summarize our technique overview. In Section 3, we analyze the streaming Kernel PCA algorithm and reach a theoretical result. In Section 4, we make a conclusion.

## 2. Technique Overview

Here in this section, we give an overview of the techniques used for our algorithm design. In general, our algorithm combines the Oja's streaming PCA algorithm [45] and a new analysis of applying kernel functions in it.

### 2.1. Streaming PCA

Our first technique is based on the Oja's traditional scheme used for streaming PCA problems. The algorithm is based on the Hebbian learning rule, which states that the connection strength between two neurons should be increased if their activity is correlated. In the context of PCA, the algorithm updates the principal component (PC) vector in the direction of the current data point, but with a learning rate that decreases over time. The algorithm aims to make the PC vector converge to the primary eigenvector of the covariance matrix of the data. This eigenvector corresponds to the direction in which the data displays the most significant variation. By utilizing this method, it becomes feasible to identify any shifts in the data distribution with time. Formally, when the data structure receives a stream of data points

$$x_1, \ldots, x_n \in \mathbb{R}^d,$$

it iteratively updates a vector $v \in \mathbb{R}^n$ (Starting from a random Gaussian vector) such that

$$v_i = v_{i-1} + \eta \cdot x_i x_i^\top v_{i-1},$$

where $\eta \in \mathbb{R}$ is the learning rate. Finally, the data structure outputs a vector

$$v_n = \prod_{i=1}^n (I_n + \eta x_i x_i^\top) v_0,$$

where $v_0 \sim \mathcal{N}(0, I_n)$. It is known that, with high probability, this output vector is close to the top principal component.

### 2.2. Applying kernel function to stream PCA

Oja's original streaming algorithm only supports traditional linear PCA questions. We want to generalize it to supporting kernel function. To do this, we need to overcome several barriers:

- **Where to apply the kernel function?** As we describe before, we need to "map" the input data points onto some "kernel" space. But for the streaming setting, how to deal with the data stream (different from the offline algorithm) becomes a question.

- **Can streaming algorithm work with kernel method?** As the classic streaming PCA algorithms mostly work for linear PCA problems. It might have several unexpected barriers to applying the kernel method here.

To overcome these barriers, we present our streaming PCA algorithm which is generalized from Oja's algorithm. To be specific, given a kernel function $\phi : \mathbb{R}^d \to \mathbb{R}^m$, our algorithm receives a stream of data points

$$x_1, \ldots, x_n \in \mathbb{R}^d.$$

It first generates a random Gaussian vector $v_0 \in \mathbb{R}^m$ at the beginning of the procedure, then it iteratively updates a vector

$$v_i = v_{i-1} + \eta \cdot \langle \phi(x_i), v_{i-1} \rangle \cdot \phi(x_i),$$

where $\eta \in \mathbb{R}$ is the learning rate. When the algorithm stops, it outputs a vector

$$v_n = \prod_{i=1}^{n}(I_n + \eta \cdot \phi(x_i)^\top \phi(x_i)) \cdot v_0.$$

By an analysis of the algorithm, we will show that, with a high probability, this vector $v_n$ is close to the top principal component as desired in Theorem 1.1.

## 2.3. Eigenvalue Ratio Implies Existence of Algorithm

In the traditional (linear) streaming PCA algorithm, it has been shown that the speed, at which the maintained vector approaches the dominant eigenvector, is determined by the relationship between the largest and second largest eigenvalues. To be specific, if $\lambda_1$ and $\lambda_2$ are the top-2 eigenvalues of the covariance matrix, we define $R := \frac{\lambda_1}{\lambda_2}$ to be the ratio of them. Let $\epsilon \in (0, 0.1)$ be an error parameter, one has the guarantee that

$$1 - \langle v_n, v^* \rangle^2 = \sin^2(v_n, v^*) \le \epsilon$$

after $O(\log_R(\frac{d}{\epsilon}))$ iterations.

In our kernel setting, we give the first analysis of this convergence result on the streaming PCA algorithm. We show that when $R \ge C \cdot \log n \cdot \log d$, modified Oja's algorithm (added kernel trick to it) provides an $\epsilon$-solution to the PCA problem. By choosing $m$ to be sufficiently large, we can increase $R$. Intuitively, as $m$ grows, the first dimension captures more information, while the second dimension captures less information.

## 2.4. Overview of Our Analysis Approach

Our analysis approach can be summarized in the following paragraphs. Our proof outline is mainly followed from [53], while we apply kernel functions in different stages of the algorithm and analysis.

**Properties Implied by Update Rule.** By the update rule of our algorithm, i.e.,

$$v_i = v_{i-1} + \eta x_i x_i^\top v_{i-1},$$

we first show the maintained vector has several simple but useful properties holding (See Claim D.1 for detailed statement and proofs), which provide the foundation for the further analysis. For example, we show that the norm of the vector continues to grow in the iterative maintenance, i.e.,

$$\|v_i\|_2^2 \ge \|v_{i-1}\|_2^2$$

for any $i \in [n]$, which (described in the next paragraph) is very useful, since the bound of the error involves an inverse proportional term of the norm of the final vector. The analysis in [53] gives proof that under a traditional setting (without kernel function), the growth of the norm is lower bounded. We follow their approach and prove a kernel version, that is, we show

$$\log(\|v_b\|_2^2 / \|v_a\|_2^2) \ge \eta \sum_{i=a+1}^{b} \langle \phi(x_i), \widehat{v}_{i-1} \rangle^2.$$

These properties are crucial in the correctness proofs, which are described in the later paragraphs.

**Never-far-away property.** As mentioned before, our algorithm iteratively maintains a vector $v_i$ such that it will converge to the top eigenvector $v^*$ of the covariance matrix (i.e., the top principal component). There is a concern about the convergence and robustness of the algorithm that, when the stream comes in an adversarial way, e.g., it puts several data points in some special directions, can our algorithm still have the convergence guarantee? Starting from this, [53] provided an approach showing that, no matter where the maintaining starts from, once the maintained vector ever gets close to the target $v^*$, it can never be *too far away* from it. We give a more detailed analysis, showing this holds even with the kernel function. Formally, we define

$$P := I - v^* v^{*\top} \in \mathbb{R}^{d \times d},$$

then for any $v_0$ and $i$, we have the result that,

$$\|P\widehat{v}_i\|_2 \leq \sqrt{\alpha} + \|Pv_0\|_2/\|v_i\|_2,$$

for some constant $\alpha$. Since our data structure has a zero-memory ability that, at some point $i$, the future output of it only depends on the current state $v_i$, and has nothing to do with the past $v_j$'s (for $j < i$), it implies the property that, if it ever gets close to the target, it will never get too far away. We call it "never-far-away" property. This result also implies that the final output will be better as the growth of the $\ell_2$ norm of the maintained vector $\|v_i\|_2$. This property is formally stated in Lemma 2.1.

**Bound on Sequence.** By Lemma 2.1, we show that if one ever gets close to $v^*$, it will never move by more than $\sqrt{\alpha}$ from it. Based on that, we further show that one cannot even move $\sqrt{\alpha}$ without increasing the norm of $v$, i.e., we show in Lemma 2.2 that if $v_0 = v^*$, for any two steps $0 \leq a \leq b \leq n$, it holds that

$$\|P\widehat{v}_b - P\widehat{v}_a\|_2^2 \leq 50 \cdot \alpha \cdot \log(\|v_b\|_2/\|v_a\|_2).$$

By the above analysis, we have the result that, to make the final output close to the desired target, one needs to make $\|v_i\|_2$ large. We first notice that, when $v_i$ drifts from the desired directions we want it to be, it can cause the reduction on $\|v_i\|_2$, i.e.,

$$\|v_i\|_2 \geq \exp(\sum_{j \in [i]} \eta \langle \phi(x_j), \widehat{v}_{j-1} \rangle^2).$$

We want to make sure that, the influence of each term $\eta \langle \phi(x_j), \widehat{v}_{j-1} \rangle$ on $\|v_i\|$ is small enough so that, the final norm of $v_N$ is large enough. So we show the following decomposition

$$\langle \phi(x_j), \widehat{v}_{j-1} \rangle^2 \geq \frac{1-\alpha}{2} \langle \phi(x_j), v^* \rangle^2 - \langle \phi(x_j), P\widehat{v}_{j-1} \rangle^2.$$

Thus, it suffices to show the second term is small enough so that it won't destroy the growth of the norm. Formally, we need prove that if $v_0 = v^*$, then for all $i \in [N]$, it holds that

$$\eta \sum_{i=1}^{n} \langle \phi(x_i), P\widehat{v}_{i-1} \rangle^2 \leq 100 \cdot \alpha^2 \cdot \log^2 n \cdot \log \|v_n\|_2.$$

As the analysis before, this implies that, if the vector maintained ever gets close to the target eigenvector, the sum of the products will be bounded, so that the norm will continue to grow. The formal statement is Lemma 2.3.

**Lower Bound.** In [53], they provided a lower bound for the norm of the output vector. We generalize their method by applying the kernel function here. The next step of our poof is to lower bound the norm of the final output. Our approach is described as follows. We first prove that the properties in Claim D.1 imply the result of lower bound on $\|v_n\|_2$. We show in Lemma 2.5 that,

$$\|v_n\|_2 \geq \sqrt{\eta} \cdot (\sum_{i \in [n]} \langle \phi(x_i), v_{i-1} \rangle^2)^{1/2}.$$

Combining this together with Lemma 2.3 we show that

$$\log(\|v_n\|_2) \geq \frac{\eta \sum_{i \in [n]} \langle v^*, \phi(x_i) \rangle^2}{8 + 8 \cdot C \cdot \alpha^2 \log^2 n},$$

which provides the lower bound for the norm of the output vector. The formal proof can be found in Lemma 2.4.

## 2.5. Analysis of Our Kernel PCA Algorithm

In this section, we provide the lemmas that are useful for our kernel PCA algorithm analysis.

**Lemma 2.1** (Growth implies correctness). *For any $v_0$ and all $i \in [n]$, we have $\|P\hat{v}_i\|_2 \leq \sqrt{\alpha} + \|Pv_0\|_2/\|v_i\|_2$. Further, if $v_0 = v^*$, then we have $\|P\hat{v}_i\|_2 \leq \sqrt{\alpha}$.*

*Proof.* See Appendix E.1 for detailed proof. □

**Lemma 2.2.** *Suppose $v_0 = v^*$. For any two time steps $0 \leq a < b \leq n$,*

$$\|P\hat{v}_b - P\hat{v}_a\|_2^2 \leq 50 \cdot \alpha \log(\|v_b\|_2/\|v_a\|_2).$$

*Proof.* See Appendix E.2 for detailed proof. □

**Lemma 2.3.** *If $v_0 = v^*$, then for $i \in [n]$, we have*

$$\eta \sum_{i=1}^{n} \langle \phi(x_i), P\hat{v}_{i-1} \rangle^2 \leq 100 \cdot \alpha^2 \cdot \log^2 n \cdot \log \|v_n\|_2.$$

*Proof.* See Appendix E.4 for detailed proof. □

**Lemma 2.4** (The right direction grows.). *Let $\alpha \in (0, 0.1)$. Let $C_1 \geq 200$ denote some fixed constant. Then if $v_0 = v^*$ we have*

$$\log(\|v_n\|_2) \geq \frac{\beta/8}{1 + C_1 \cdot \alpha^2 \log^2 n}.$$

*Further, if $\alpha \in (0, 1/(10C_1 \log n))$, we have*

$$\|v_n\|_2 \geq \exp(\beta/20).$$

*Proof.* See Appendix E.5 for detailed proof. □

**Lemma 2.5.** *We have $\|v_n\|_2 \geq \sqrt{\eta} \cdot (\sum_{i=1}^{n} \langle \phi(x_i), v_{i-1} \rangle^2)^{1/2}$*

*Proof.* See Appendix E.6 for detailed proof. □

## 3. Our Kernel PCA Result

In this section, we show our results for the kernel PCA algorithm. In Section 3.1, we provide a guarantee for the final output. In Section 3.2, we formally present the main result of our streaming algorithm.

### 3.1. The Guarantee of Final Output

**Theorem 3.1.** *Let $C \geq 10^3$ be a sufficiently large constant. Suppose that $\alpha \in (0, \frac{1}{C \log n})$ and $\beta \geq C \log d$. Our algorithm outputs a vector $\hat{v}_n \in \mathbb{R}^d$ such that*

$$\Pr[\|P\hat{v}_n\|_2 \leq \sqrt{\alpha} + \exp(-\beta/200)] \geq 1 - \exp(-\beta/200)$$

*Proof.* Our algorithm starts with a uniform random direction $\hat{v}_0$, and the sequence of $\hat{v}_i$ doesn't depend on $\|v_0\|_2$, so we can assume $v_0 \sim \mathcal{N}(0, I_d)$.

By this assumption, we know that for each $i \in [d]$, $(v_0)_i \sim \mathcal{N}(0, 1)$. Hence, we sum over all the initial vectors $v_0$ for the sequence of $\hat{v}_i$ to get

$$\mathbb{E}[\|v_0\|_2^2] = \sum_{i=1}^{d} \mathbb{E}[\|(v_0)_i\|_2^2] = \sum_{i=1}^{d} 1 = d$$

where the first step follows from our assumption for proof, and the second step follows from the definition of Gaussian.

We define vector $v_0 \in \mathbb{R}^d$ as $v_0 := a \cdot v^* + u$ for $u \perp v^*$ and $a \sim \mathcal{N}(0, 1)$.

We define matrix $B \in \mathbb{R}^{d \times d}$

$$B := \prod_{i=1}^{n} (1 + \eta \cdot \phi(x_i) \cdot \phi(x_i)^\top),$$

so by Definition C.4 (update rule), $v_n = Bv_0$.

With probability $1 - \delta$, we get

$$\|v_n\|_2 = \|Bv_0\|_2 = \|aBv^* + Bu\|_2 \geq \delta \cdot \|Bv^*\|_2 \geq \delta \cdot \exp(\beta/20) \tag{1}$$

where the first step follows from $v_n = Bv_0$, the second step follows from $v_0 = av^* + u$, the third step follows from Claim B.8, and the last step follows from Lemma 2.4.

We can compute expectation,

$$
\begin{aligned}
\mathbb{E}[\|u\|_2^2] &= \mathbb{E}[\|v_0\|_2^2 - \|av^*\|_2^2 - 2\langle av^*, u \rangle] \\
&= \mathbb{E}[\|v_0\|_2^2] - \mathbb{E}[\|av^*\|_2^2] - \mathbb{E}[2\langle av^*, u \rangle] \\
&= d - \mathbb{E}[\|av^*\|_2^2] - \mathbb{E}[2\langle av^*, u \rangle] \\
&= d - 1 - \mathbb{E}[2\langle av^*, u \rangle] \\
&= d - 1
\end{aligned}
$$

where the first step follows from our definition for proof that $v_0 := a \cdot v^* + u$, the second step follows from simple algebra, the third step follows from definition of Gaussian, the fourth step follows from $\mathbb{E}[a^2] = 1$ and $\|v^*\|_2^2 = 1$, the last step follows from $\langle u^*, u \rangle = 0$.

Then applying Lemma B.6, we will have

$$\Pr[\|u\|_2^2 \geq d/\delta] \leq \mathbb{E}[\|u\|_2^2]/(d/\delta) = (d-1)\frac{\delta}{d} \leq \delta \tag{2}$$

the last step follows from $(d-1)/d \leq 1$.

The above equation implies

$$\Pr[\|u\|_2 \leq \sqrt{d/\delta}] \geq 1 - \delta.$$

With probability $1 - 3\delta$, we have

$$
\begin{aligned}
\|P\widehat{v}_n\|_2 &\leq \sqrt{\alpha} + \frac{\|u\|_2}{\|v_n\|_2} \\
&\leq \sqrt{\alpha} + \frac{\sqrt{d/\delta}}{\|v_n\|_2} \\
&\leq \sqrt{\alpha} + \frac{\sqrt{d/\delta}}{\delta \cdot \exp(\beta/20)} \\
&\leq \sqrt{\alpha} + 8 \cdot \sqrt{d} \cdot \exp(-\beta/30) \\
&\leq \sqrt{\alpha} + \exp(-\beta/40) \\
&\leq \sqrt{\alpha} + \exp(-\beta/200)
\end{aligned}
$$

where the first step follows from Lemma 2.1, and the second step follows from Eq.(2), the third step follows from Eq.(1), and the fourth step follows from choosing $\delta = \exp(-\beta/200)/4$, and the fifth step follows from $\beta \geq C \log d$ with $C \geq 500$.

$\square$

## 3.2. Main Result

**Theorem 3.2** (Formal version of Theorem 1.1). *Let $\phi : \mathbb{R}^d \to \mathbb{R}^m$. Let $\Sigma = \frac{1}{n} \sum_{i=1}^{n} \phi(x_i)\phi(x_i)^\top \in \mathbb{R}^{m \times m}$. We define $R := \lambda_1(\Sigma)/\lambda_2(\Sigma)$ where $\lambda_1(\Sigma)$ is the largest eigenvalue of $\Sigma$ and $\lambda_2(\Sigma)$ is the second*

*largest eigenvalue of $\Sigma$. Let $x^*$ denote the top eigenvector of $\Sigma$. Let $C > 10^3$ denote a sufficiently large constant. If $R \geq C \cdot (\log n) \cdot (\log d)$, there is a streaming algorithm (Algorithm 1) that only uses $O(d + m)$ spaces and receives $x_1, x_2, \cdots, x_n$ in the online/streaming fashion, and outputs a unit vector $u$ such that $1 - \langle x^*, u \rangle^2 \leq (\log d)/R$ holds with probability at least $1 - \exp(-\Omega(\log d))$.*

*Proof.* Let $C \geq 10^3$ be a sufficiently large constant. Suppose that $\alpha \in (0, \frac{1}{C \log n})$ and $\beta \geq C \log d$.

From Theorem 3.1, we have $\|Pu\|_2 \leq \epsilon$ where $\epsilon = \sqrt{\alpha} + \exp(-\beta/200)$. .

Using Claim B.2, we know that $1 - \langle u, x^* \rangle^2 \leq \epsilon^2$. From our assumption for proof, we have

$$R \geq C \cdot (\log n) \cdot (\log d) \geq \frac{1}{4} C \cdot (\log n) \cdot (\log d) \tag{3}$$

where the second step follows from $C \cdot (\log n) \cdot (\log d) \geq 0$.

Rewriting Eq. (3), we get $\frac{1}{4}(\log d)/R \leq \frac{1}{C \log n}$.

Hence, we can choose

$$\alpha = \frac{1}{4}(\log d)/R \tag{4}$$

by its domain $\alpha \in (0, \frac{1}{C \log n})$.

Eq. (4) equivalently yields that $\sqrt{\alpha} = \frac{1}{2}\sqrt{(\log d)/R}$.

Since $R \geq 1$ by the definition and we choose $\beta \geq C \log(R/(\log d))$, then

$$\begin{aligned}
\exp(-\beta/200) &\leq \exp(-(C/200)\log(R/\log d)) \\
&\leq ((\log d)/R)^2 \\
&\leq \frac{1}{2}\sqrt{(\log d)/R}.
\end{aligned}$$

where the second step follows from $C/200 \geq 4$, the last step follows from $R \geq 4 \log d$.

Thus, we have

$$\epsilon \leq \frac{1}{2}\sqrt{(\log d)/R} + \frac{1}{2}\sqrt{(\log d)/R} = \sqrt{(\log d)/R},$$

where the first step follows from $\epsilon = \sqrt{\alpha} + \exp(-\beta/200)$.

By taking square on both sides, the above implies that

$$\epsilon^2 \leq (\log d)/R.$$

So, the overall condition, we choose for $\beta$ is

$$\beta \geq C \cdot (\log d + \log(R/\log d)).$$

From Eq. (4), we knew $R$ has to satisfy that

$$R \geq (C/4) \log n \cdot \log d.$$

The failure probability is at most

$$\exp(-\beta/200) \leq \exp(-(C/200)\log(d) - \log((C/4)\log n)) \leq \exp(-\Omega(\log d)).$$

Therefore, we conclude that the probability, where the condition $1 - \langle x^*, u \rangle^2 \leq (\log d)/R$ holds, is at least $1 - \exp(-\Omega(\log d))$ as expected. $\qquad\square$

# 4. Conclusion

In conclusion, our study presents a groundbreaking streaming algorithm for kernel Principal Component Analysis (PCA), notable for its minimal space requirement of only $O(m + d)$, where $m$ is the dimension of the kernel space, and $d$ is the dimension of each data point in the dataset. This marks a significant improvement in efficiency and resource management, particularly in handling large datasets common in modern data analysis scenarios. Our algorithm, building on the foundation of Oja's traditional scheme, not only extends its application to kernel PCA but also enhances its adaptability and effectiveness in a wider range of data structures.

The core of our contribution lies in the detailed conditions we provide for the algorithm's optimal performance, particularly concerning the ratio of top eigenvectors. This insight is critical for practitioners and researchers, guiding the effective application of our algorithm in diverse scenarios. Moreover, this aspect of our work underscores the algorithm's robustness and reliability, ensuring its utility in practical, real-world data analysis tasks in fields such as web-related applications and so on.

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

# Appendix

**Roadmap.** In Section A, we provide more literature related to our paper. In Section B, we provide preliminary. In Section C we present the statements which are useful to prove the main Theorem 1.1. In Section D, we provide proof of some properties implied by the Update Rule. In Section E, we provide more proof and analysis of the Kernel PCA Algorithm.

## A. More Related Work

**Large Scale Optimization.** Principal component analysis and its kernel variant can be applied to large-scale optimization tasks as a dimensionality reduction technique to improve the efficiency of high-dimensional computations. Diffusion models [54], as well as it's high order variant [55] are a class of generative models that iteratively refine data through a diffusion process of noise addition and removal, effectively performing a large-scale optimization of the data distribution [56]. Flow matching [57] is a technique for training continuous normalizing flow models by aligning probability flow trajectories, offering an alternative paradigm for large-scale distribution alignment in generative modeling [58–60]. On the other hand, transformer-based neural networks [61] have rapidly emerged as the dominant architecture for natural language processing in machine learning. When expanded to billions of parameters and trained on vast, diverse datasets, these systems are typically termed large language models (LLMs) or foundation models [62]. Prominent LLM examples encompass BERT [63], PaLM [64], Llama [65, 66], and GPT4o [67], which display adaptable competencies [68] across numerous downstream applications. To enhance LLMs for domain-specific uses, researchers have created multiple adaptation approaches. These include: adapter modules [69–72]; calibration mechanisms [73–75]; multitask refinement [76–79]; along with prompt engineering [80, 81], scratchpad approaches [82], instruction optimization [83–85], symbolic adaptation [86–88], black-box adjustments [89], human-aligned reinforcement learning [90, 91], and structured reasoning techniques [92–95]. Contemporary investigations cover tensor architecture innovations [96–99], efficiency enhancements [100, 101, 101–123], plus ancillary studies [60, 124–157]. There are also some method devote to use model compression to improve the efficiency and deployment of LLMs [158] for its effectiveness in reducing computational overhead while preserving performance. Common compression techniques include quantization [159–161], pruning [103, 115, 162–172], and knowledge distillation [173–177]. Specifically, pruning techniques have been developed extensively, such as unstructured pruning, which removes individual weights [115, 168], and structured pruning, which eliminates entire components like neurons or attention heads [172, 178, 179]. The attention mechanism has faced criticism due to its quadratic time complexity with respect to context length [61]. Addressing this criticism, a variety of approaches are employed, including sparse attention [115, 164, 167, 180–183], low-rank approximations [69, 125, 184–187], and kernel-based methods [188–191], to reduce computational overhead and improve scalability. [192] enable the derivation of a low-rank representation of the attention matrix, which accelerates both the training and inference processes of single attention layer, tensor attention, and multi-layer transformer, achieving almost linear time complexity [32, 96, 97, 108, 157, 193, 194]. Other approaches like Mamba [195, 196], Linearizing Transformers [197, 198], Hopfield Models [104–106, 109, 121–123, 199], and PolySketchFormer [200] focus on architectural modifications and implementation optimizations to enhance performance. System-level optimizations such as FlashAttention [201–203] and block-wise parallel decoding [204] further improve efficiency. Collectively, these innovations have significantly augmented transformer models' ability to handle longer input sequences, unlocking broader applications across multiple sectors [78, 113, 116, 117, 205–210].

## B. Preliminary

We provide notations in Section B.1. We state some basic algebra and probability tools in Section B.2 and Section B.3 respectively.

## B.1.  Notations

For a matrix $A$, we use $A^\top$ to denote its transpose. For a square matrix $A$, we use $\mathrm{tr}[A]$ to denote its trace. For a vector $x \in \mathbb{R}^n$, we use $\|x\|_2$ to denote its $\ell_2$ norm, i.e., $\|x\|_2 := (\sum_{i=1}^n x_i^2)^{1/2}$.

We say a square matrix $P \in \mathbb{R}^{d \times d}$ is a projection matrix if $P^2 = P$.

For two functions $f, g$, we use the shorthand $f \lesssim g$ (resp. $\gtrsim$) to indicate that $f \leq Cg$ (resp. $\geq$) for an absolute constant $C$. We use $f \eqsim g$ to mean $cf \leq g \leq Cf$ for constants $c > 0$ and $C > 0$.

For a function $h(j)$ with its domain $X$, we use $\arg\max_{j \in X} h(j)$ to denote the corresponding index $j$ for the largest output of function $h(j)$.

We use $\mathbb{E}[\cdot]$ to denote the expectation, and $\Pr[\cdot]$ to denote the probability.

For a distribution $D$ and a random variable $x$, we use $x \sim D$ to denote that we draw a random variable from the distribution $D$.

We use $\mathcal{N}(\mu, \sigma^2)$ to denote a Gaussian distribution with mean $\mu$ and variance $\sigma^2$.

For arbitrary functions $f(x) \in \mathbb{R}$ and $g(x) \in \mathbb{R}$, if $\exists M \in \mathbb{R}^+$ and $x_0 \in \mathbb{R}$, such that $|f(x)| \leq M \cdot g(x)$ for all $x > x_0$. We denote that $f(x) = O(g(x))$.

For arbitrary functions $f(x) \in \mathbb{R}$ and $g(x) \in \mathbb{R}$, if $\exists k \in \mathbb{R}^+$ and $x_1 \in \mathbb{R}$, such that $|f(x)| \geq k \cdot g(x)$ for all $x > x_1$. We denote that $f(x) = \Omega(g(x))$.

For arbitrary functions $f(x) \in \mathbb{R}$ and $g(x) \in \mathbb{R}$, if $f(x) = O(g(x))$ and $f(x) = \Omega(g(x))$, we denote that $f(x) = \Theta(g(x))$.

**Definition B.1.** *Let $\phi : \mathbb{R}^d \to \mathbb{R}^m$ denote a kernel function. We define $\Sigma := \frac{1}{n} \sum_{i=1}^n \phi(x_i)\phi(x_i)^\top$.*

## B.2.  Basic Algebra Tools

**Claim B.2.** *Let $P = (I - v^*(v^*)^\top)$ where $P \in \mathbb{R}^{d \times d}$. Let $u \in \mathbb{R}^d$ denote any unit vector $\|u\|_2 = 1$, if $\|Pu\|_2 \leq \epsilon$, then have*

$$1 - \langle u, v^* \rangle^2 \leq \epsilon^2.$$

*Proof.*  We have

$$
\begin{aligned}
\epsilon^2 &\geq \|Pu\|_2^2 \\
&= u^\top P P u \\
&= u^\top P u \\
&= u^\top u - u^\top v^*(v^*)^\top u \\
&= 1 - \langle u, v^* \rangle^2
\end{aligned}
$$

where the first step follows from our assumption for proof, the second step follows from the property of norm, the third step follows from the definition of projection matrix $P^2 = P$, the fourth step follows from our definition for proof that $P = (I - v^*(v^*)^\top)$, and the last step follows from $a^\top b = \langle a, b \rangle$. $\qquad\square$

**Fact B.3.** *For any integer $A$, and integer $k$, we define $f_k := \lfloor A/2^k \rfloor$ and $f_{k+1} := 2 \cdot \lfloor A/2^{k+1} \rfloor$. Then, we have*

$$|f_k - f_{k+1}| \leq 1$$

*Proof.*  We can always write $A$

$$A = B \cdot 2^{k+1} + C \cdot 2^k + D$$

where $B \geq 0$, $C \in \{0, 1\}$, and $D \in [0, 2^k - 1]$.

We have

$$|f_k - f_{k+1}| = |(2B + C) - 2B| = C \le 1$$

Thus, we complete the proof.

$\square$

**Claim B.4.** *Let $0 \le a_1, a_2, ..., a_n$. For each $i \in \{0, 1, \cdots, n\}$, we define*

$$b_i := \exp(\sum_{j=0}^{i} a_j)$$

*for $i \in \{0, 1, ..., n\}$.*

*Then:*

$$\sum_{i=1}^{n} a_i b_{i-1} \le b_n.$$

*Proof.* This follows from induction on $n$. $n = 0$ is trivial, and then for $k \in \{0, 1, ..., n\}$ and $k < n$, we have the following case for $k + 1 \in \{0, 1, ..., n\}$.

$$\sum_{i=1}^{k+1} a_i b_{i-1} \le b_k + a_{k+1} b_k$$
$$= (1 + a_{k+1}) b_k$$
$$\le e^{a_{k+1}} b_k$$
$$\le b_k,$$

where the first step follows from the induction, the second step follows from multiplicative distribution, the third step follows from the Maclaurin Series of the exponential function, and the last step follows from our definition for proof. $\square$

**Claim B.5.** *For any $x \in \mathbb{R}, y \in \mathbb{R}$, we have*

$$(x + y)^2 \ge \frac{1}{2}x^2 - y^2.$$

*Proof.* It's equivalent to

$$x^2 + 2xy + y^2 \ge \frac{1}{2}x^2 - y^2,$$

which is equivalent to

$$\frac{1}{2}x^2 + 2xy + 2y^2 \ge 0,$$

which is further equivalent to

$$\frac{1}{2}(x + 2y)^2 \ge 0.$$

Thus, we complete the proof.

$\square$

## B.3. Basic Probability Tools

**Lemma B.6** (Markov's inequality). *If $X$ is a non-negative random variable and $a > 0$, then*

$$\Pr[X \ge a] \le \mathbb{E}[X]/a.$$

**Lemma B.7** (Anti-concentration of Gaussian distribution, see Lemma A.4 in [211] for an example)**.**
*Let $X \sim \mathcal{N}(0, \sigma^2)$, that is the probability density function of $X$ is given by*

$$\phi(x) = \frac{1}{\sqrt{2\pi\sigma^2}} \exp(-x^2/(2\sigma^2)).$$

*Then*

$$\frac{2}{3}t/\sigma \leq \Pr[|X| \leq t] \leq \frac{4}{5}t/\sigma.$$

**Claim B.8.** *Let $a \sim \mathcal{N}(0, 1)$.*

*For any two vectors $u \in \mathbb{R}^d$ and $v \in \mathbb{R}^d$, then we have*

$$\Pr_{a \sim \mathcal{N}(0,1)}[\|au + v\|_2 \geq \delta\|u\|_2] \geq 1 - \delta.$$

*Proof.* We define

$$x := \|au + v\|_2^2.$$

**Case 1.** There exists some scalar $b \in \mathbb{R}$ such that $v = b \cdot u$.

Then we have

$$x = (a + b)^2\|u\|_2^2.$$

Recall that the goal of this lemma is to prove

$$\Pr_{a \sim \mathcal{N}(0,1)}[\sqrt{x} \geq \delta\|u\|_2] \geq 1 - \delta.$$

It is equivalent to

$$\Pr_{a \sim \mathcal{N}(0,1)}[x \geq \delta^2\|u\|_2^2] \geq 1 - \delta.$$

Using the Equation of $x = (a + b)^2\|u\|_2^2$, the statement is equivalent to

$$\Pr_{a \sim \mathcal{N}(0,1)}[(a + b)^2\|u\|_2^2 \geq \delta^2\|u\|_2^2] \geq 1 - \delta,$$

which is equivalent to

$$\Pr_{a \sim \mathcal{N}(0,1)}[(a + b)^2 \geq \delta^2] \geq 1 - \delta.$$

By the property of Gaussian, we know that

$$\Pr_{a \sim \mathcal{N}(0,1)}[(a + b)^2 \geq \delta^2] \geq \Pr_{a \sim \mathcal{N}(0,1)}[(a + 0)^2 \geq \delta^2].$$

Thus, we just need to show that

$$\Pr_{a \sim \mathcal{N}(0,1)}[a^2 \geq \delta^2] \geq 1 - \delta.$$

The above equation directly follows from Lemma B.7.

**Case 2.** There exists some scalar $b$ and vector $w$ such that $\langle u, w \rangle = 0$ and

$$v = b \cdot u + w.$$

In this case,

$$\begin{aligned}
x &= \|(a + b)u + w\|_2^2 \\
&= (a + b)^2\|u\|_2^2 + \|w\|_2^2 \\
&> (a + b)^2\|u\|_2^2.
\end{aligned}$$

The remaining of the proof is identical to case 1, since $x$ is becoming larger now. $\square$

# C. Basic Definitions Properties of Streaming Kernel PCA Algorithm and Update Rules

In Section C.1, we define sample vectors for Kernel PCA analysis. In Section C.2, we provide an update rule for our streaming algorithm.

## C.1. Definitions of Vectors

We formally define $\alpha, \eta > 0$ and $v^* \in \mathbb{R}^d$ and $\beta > 0$ as follows:

**Definition C.1.** *Let $\beta$ and $\alpha$ denote two parameters that $\beta \geq \alpha > 0$.*

*For each $i \in [n]$, we use $x_i \in \mathbb{R}^d$ to denote the sample. Let $\eta \in (0, 0.1)$ be the learning rate.*

*We define vectors $v^* \in \mathbb{R}^d$ as follows:*

- $\|v^*\|_2 = 1$,

- $\eta \sum_{i=1}^n \langle v^*, \phi(x_i) \rangle^2 = \beta$,

- *for all vectors $w$ with $\|w\|_2 \leq 1$ and $\langle w, v^* \rangle = 0$, we have $\eta \sum_{i=1}^n \langle w, \phi(x_i) \rangle^2 \leq \alpha$.*

Without loss of generality, we keep $\|v^*\|_2 = 1$ for the entire algorithm analysis. We define our projection operator based on $v^*$.

**Definition C.2.** *We define $P = I - v^*(v^*)^\top$ to be the projection matrix that removes the $v^*$ component.*

We have the following claim.

**Claim C.3.** *Since $P = I - v^*(v^*)^\top$ and $\|v^*\|_2 = 1$, then we have*

$$Pv^* = 0.$$

## C.2. Update Rule

**Definition C.4.** *Let $\eta$ denote some parameters. We define an updated rule as follows:*

$$v_i := v_{i-1} + \eta \langle \phi(x_i), v_{i-1} \rangle \phi(x_i).$$

*Then, we can rewrite it as*

$$v_i = (I + \eta \phi(x_i) \phi(x_i)^\top) v_{i-1}.$$

For stability, an implementation would only keep track of the normalized vectors $\widehat{v}_i = v_i / \|v_i\|_2$. For analysis purposes, we will often consider the unnormalized vectors $v_i$.

**Definition C.5.** *Let $v_i$ denote the unnormalized vectors, for all $i \in [n]$. We define $\widehat{v}_i$ as follows*

$$\widehat{v}_i := v_i / \|v_i\|_2.$$

# D. Proof of The Properties Implied by Update Rule

**Claim D.1.** *For any parameter $\eta > 0$. By relationship between $v_i$ and $v_{i-1}$ (see Definition C.4), we have*

- *Property 1.*

$$\|v_i\|_2^2 = \|v_{i-1}\|_2^2 \cdot (1 + (2\eta + \eta^2 \|\phi(x_i)\|_2^2) \cdot \langle \phi(x_i), \widehat{v}_{i-1} \rangle^2)$$

- *Property 2.*

$$\|v_i\|_2^2 \geq \|v_{i-1}\|_2^2, \forall i \in [n]$$

- *Property 3. If we additionally assume $\eta \leq 0.1/\max_{i \in [n]} \|\phi(x_i)\|_2^2$,*

$$\log(\|v_i\|_2^2/\|v_{i-1}\|_2^2) \geq \eta\langle\phi(x_i), \widehat{v}_{i-1}\rangle^2.$$

- *Property 4.*

$$\log(\|v_b\|_2^2/\|v_a\|_2^2) \geq \sum_{i=a+1}^{b} \eta\langle\phi(x_i), \widehat{v}_{i-1}\rangle^2$$

- *Property 5. For any integers $b > a$*

$$v_b - v_a = \sum_{i=a+1}^{b} \eta\phi(x_i)\phi(x_i)^\top v_{i-1}$$

*Proof.* **Proof of Property 1.**

Recall Definition C.4, we have

$$v_i = v_{i-1} + \eta \cdot \langle\phi(x_i), v_{i-1}\rangle\phi(x_i).$$

Taking the norm square on both sides of the above equation, we have

$$\|v_i\|_2^2 = \|v_{i-1}\|_2^2 + 2\eta \cdot \langle\phi(x_i), v_{i-1}\rangle\langle v_{i-1}, \phi(x_i)\rangle + \eta^2 \cdot \langle\phi(x_i), v_{i-1}\rangle^2\|\phi(x_i)\|_2^2.$$

We rewrite it as

$$\begin{aligned}
\|v_i\|^2 &= \|v_{i-1}\|_2^2 + 2\eta\langle\phi(x_i), v_{i-1}\rangle\langle v_{i-1}, \phi(x_i)\rangle + \eta^2\langle\phi(x_i), v_{i-1}\rangle^2\|\phi(x_i)\|_2^2 \\
&= \|v_{i-1}\|_2^2 + 2\eta\langle\phi(x_i), v_{i-1}\rangle^2 + \eta^2\langle\phi(x_i), v_{i-1}\rangle^2\|\phi(x_i)\|_2^2 \\
&= \|v_{i-1}\|_2^2 + 2\eta\langle\phi(x_i), \widehat{v}_{i-1}\rangle^2 \cdot \|v_{i-1}\|_2^2 + \eta^2\langle\phi(x_i), \widehat{v}_{i-1}\rangle^2 \cdot \|v_{i-1}\|_2^2\|\phi(x_i)\|_2^2 \\
&= \|v_{i-1}\|_2^2 \cdot (1 + (2\eta + \eta^2\|\phi(x_i)\|_2^2) \cdot \langle\phi(x_i), \widehat{v}_{i-1}\rangle^2)
\end{aligned}$$

where the third step follows from Definition C.5 ($\widehat{v}_{i-1} = v_{i-1}/\|v_{i-1}\|_2$).

**Proof of Property 2.** The proof of this statement is going to use Property 1 in some steps as a black-box. We first consider the terms $(2\eta + \eta^2\|\phi(x_i)\|_2^2)$ and $\langle\phi(x_i), \widehat{v}_{i-1}\rangle^2$.

For $(2\eta + \eta^2\|\phi(x_i)\|_2^2)$, we have $\|\phi(x_i)\|_2^2 \geq 0$.

By Definition C.1, we get $2\eta > 0$ and $\eta^2 > 0$. Hence,

$$2\eta + \eta^2\|\phi(x_i)\|_2^2 > 0.$$

For $\langle\phi(x_i), \widehat{v}_{i-1}\rangle^2$, it is obvious that this term is greater than or equal to 0. Thus, we have

$$\langle\phi(x_i), \widehat{v}_{i-1}\rangle^2 \geq 0.$$

Therefore, we conclude that

$$\begin{aligned}
\|v_i\|^2 &= \|v_{i-1}\|_2^2 \cdot (1 + (2\eta + \eta^2\|\phi(x_i)\|_2^2) \cdot \langle\phi(x_i), \widehat{v}_{i-1}\rangle^2) \\
&\geq \|v_{i-1}\|_2^2 \cdot (1 + 0) \\
&= \|v_{i-1}\|_2^2,
\end{aligned}$$

where the second step follows from the inequality relationship and $i \in [n]$.

**Proof of Property 3.** From property 1, we have

$$\frac{\|v_i\|_2^2}{\|v_{i-1}\|_2^2} = 1 + (2\eta + \eta^2\|\phi(x_i)\|_2^2) \cdot \langle\phi(x_i), \widehat{v}_{i-1}\rangle^2.$$

Taking the log both sides, we have

$$\log(\frac{\|v_i\|_2^2}{\|v_{i-1}\|_2^2}) = \log(1 + (2\eta + \eta^2\|\phi(x_i)\|_2^2) \cdot \langle\phi(x_i), \widehat{v}_{i-1}\rangle^2).$$

We define $u = (2\eta + \eta^2\|\phi(x_i)\|_2^2) \cdot \langle\phi(x_i), \widehat{v}_{i-1}\rangle^2$. We need to show that $u \in [0, 1.5]$.

For the lower bound case, it is obvious that $u \geq 0$ since $\eta \geq 0$.

Next, we prove the upper bound case,

$$
\begin{aligned}
u &= (2\eta + \eta^2\|\phi(x_i)\|_2^2) \cdot \langle\phi(x_i), \widehat{v}_{i-1}\rangle^2 \\
&= (2\eta + \eta^2\|\phi(x_i)\|_2^2) \cdot \|\phi(x_i)\|_2^2 \cdot \langle\phi(x_i)/\|\phi(x_i)\|_2, \widehat{v}_{i-1}\rangle^2 \\
&\leq (2\eta + \eta^2\|\phi(x_i)\|_2^2) \cdot \|\phi(x_i)\|_2^2, \\
&\leq 2 \cdot 0.1 + 0.1^2 \\
&\leq 0.3
\end{aligned}
$$

where the third step follows from $\langle a, b\rangle^2 \leq 1$ for any $\|a\|_2 = \|b\|_2 = 1$, the fourth step follows from $\eta \leq 0.1/\|\phi(x_i)\|_2^2$.

It is not hard to see that for any $u \in [0, 1.5]$

$$\log(1 + u) \geq 0.25 \cdot u.$$

Thus,

$$
\begin{aligned}
\log(1 + u) &\geq 0.25 \cdot (2\eta + \eta^2\|\phi(x_i)\|_2^2) \cdot \langle\phi(x_i), \widehat{v}_{i-1}\rangle^2 \\
&\geq 0.5\eta\langle\phi(x_i), \widehat{v}_{i-1}\rangle^2.
\end{aligned}
$$

**Proof of Property 4.** From property 3, we have

$$\log(\|v_i\|_2^2/\|v_{i-1}\|_2^2) \geq \eta\langle\phi(x_i), \widehat{v}_{i-1}\rangle^2.$$

$\forall a, b \in [n]$ and $a < b$, we have

$$
\begin{aligned}
&\log(\|v_b\|_2^2/\|v_a\|_2^2) \\
&= \log(\frac{\|v_b\|_2^2}{\|v_{b-1}\|_2^2} \cdot ... \cdot \frac{\|v_{a+1}\|_2^2}{\|v_a\|_2^2}) \\
&= \log(\frac{\|v_b\|_2^2}{\|v_{b-1}\|_2^2}) + ... + \log(\frac{\|v_{a+1}\|_2^2}{\|v_a\|_2^2}) \\
&\geq \eta\langle\phi(x_b), \widehat{v}_{b-1}\rangle^2 + ... + \eta\langle\phi(x_{a+1}), \widehat{v}_a\rangle^2 \\
&= \sum_{i=a+1}^{b} \eta\langle\phi(x_i), \widehat{v}_{i-1}\rangle^2
\end{aligned}
$$

where the second step follows from $\log(ab) = \log(a) + \log(b)$, and the third step follows from Property 3.

**Proof of Property 5.** By Definition C.4, we have $v_i = (I + \eta\phi(x_i)\phi(x_i)^\top)v_{i-1}$.

We rewrite this as

$$
\begin{aligned}
v_i - v_{i-1} &= (I + \eta\phi(x_i)\phi(x_i)^\top)v_{i-1} - v_{i-1} \\
&= \eta\phi(x_i)\phi(x_i)^\top v_{i-1},
\end{aligned} \tag{5}
$$

where the first step follows from Definition. C.4.

Then $\forall a, b \in [n]$ and $a < b$, we have

$$
\begin{aligned}
v_b - v_a &= v_b - v_{b-1} + ... + v_{a+1} + v_a \\
&= \eta\phi(x_b)\phi(x_b)^\top v_{b-1} + ... + \eta\phi(x_{a+1})\phi(x_{a+1})^\top v_a \\
&= \sum_{i=a+1}^{b} \eta\phi(x_i)\phi(x_i)^\top v_{i-1}
\end{aligned}
$$

where the second step follows from Eq. (5). $\qquad\square$

# E. Analysis of Our Kernel PCA Algorithm

Section E.1, we provide the property, growth implies correctness, of our defined vector.

Section E.2, we provide the projection operator and show the property of increasing the norm of our defined vector.

In Section E.3, we provide a bound on sequences.

In Section E.4, we provide an upper bound for the summation of the inner product.

In Section E.5, we provide a lower bound on the log of the norm of the final output by our streaming algorithm.

In Section E.6, we show the lower bound of $\ell_2$ norms of the final vector generated by our algorithm.

## E.1. Growth implies correctness

**Lemma E.1** (Restatement of Lemma 2.1). *For any $v_0$ and all $i \in [n]$, we have*

$$\|P\widehat{v}_i\|_2 \leq \sqrt{\alpha} + \|Pv_0\|_2/\|v_i\|_2.$$

*Further, if $v_0 = v^*$, then we have*

$$\|P\widehat{v}_i\|_2 \leq \sqrt{\alpha}.$$

*Proof.* We will prove this for the final index $i = n$.

Without loss of generality, we can assume $\|v_0\|_2 = 1$ over the entire proof. Then for any unit vector $w \perp v^*$,

$$\langle v_n - v_0, w \rangle$$
$$= \eta \sum_{i=1}^{n} \langle \phi(x_i), v_{i-1} \rangle \langle \phi(x_i), w \rangle$$
$$\leq \eta (\sum_{i=1}^{n} \langle \phi(x_i), v_{i-1} \rangle^2)^{1/2} \cdot (\sum_{i=1}^{n} \langle \phi(x_i), w \rangle^2)^{1/2}$$
$$\leq \|v_n\|_2 \cdot \sqrt{\eta} \cdot (\sum_{i=1}^{n} \langle \phi(x_i), w \rangle^2)^{1/2}$$
$$\leq \|v_n\|_2 \cdot \sqrt{\alpha} \tag{6}$$

where the first step follows from Property 5 of Claim D.1, the second step follows from Cauchy-Schwartz, the third step follows from Lemma E.7, and the last step follows from Definition C.1.

Hence

$$\langle \widehat{v}_n, w \rangle \leq \frac{1}{\|v\|_2} \langle v_n, w \rangle$$
$$= \frac{1}{\|v_n\|_2} (\langle v_n - v_0, w \rangle + \langle v_0, w \rangle)$$
$$\leq \sqrt{\alpha} + \frac{\langle v_0, w \rangle}{\|v_n\|_2}. \tag{7}$$

where the first step follows from the definition of $\widehat{v}_n$, the second step follows from subtracting and adding the same term, and the third step follows from Eq. (6).

Setting $w = P\widehat{v}_n/\|P\widehat{v}_n\|_2$, we have

$$\langle \widehat{v}_n, w \rangle = \langle \widehat{v}_n, P\widehat{v}_n/\|P\widehat{v}_n\|_2 \rangle$$
$$= \langle \widehat{v}_n, P^2\widehat{v}_n/\|P\widehat{v}_n\|_2 \rangle$$

$$= \widehat{v}_n^\top P^2 \widehat{v}_n / \|P\widehat{v}_n\|_2$$
$$= \|P\widehat{v}_n\|_2 \tag{8}$$

where the second step follows from $P$ is a projection matrix (which implies $P^2 = P$), the third step follows from the properties of the inner product for Euclidean vector space, and the last step follows from $a^\top B^2 a = \|Ba\|_2^2$ for any matrix $B$ and vector $a$.

We also know that

$$\langle v_0, w \rangle = \langle v_0, P\widehat{v}_n / \|P\widehat{v}_n\|_2 \rangle$$
$$= \langle Pv_0, P\widehat{v}_n / \|P\widehat{v}_n\|_2 \rangle$$
$$\leq \|Pv_0\|_2 \cdot \|P\widehat{v}_n\|_2 / \|P\widehat{v}_n\|_2$$
$$\leq \|Pv_0\|_2, \tag{9}$$

where the second step follows from $P$ is a projection matrix (which implies that $P^2 = P$), the third step follows from $\langle a, b \rangle \leq \|a\|_2 \cdot \|b\|_2$.

Now, we can conclude that

$$\|P\widehat{v}_n\|_2 = \langle \widehat{v}_n, w \rangle$$
$$\leq \sqrt{\alpha} + \frac{\langle v_0, w \rangle}{\|v_n\|_2}$$
$$\leq \sqrt{\alpha} + \|Pv_0\|_2 / \|v_n\|_2$$

where the first step follows from Eq. (8), the second step follows from Eq. (7), and the last step follows from Eq. (9).

For the case $v_0 = v^*$, since $Pv^* = 0$, we have $\|P\widehat{v}_i\|_2 \leq \sqrt{\alpha}$ as desired.

Therefore, we complete the proof. $\qquad\square$

## E.2. The Projection Operator

Using Lemma E.1, we show that if we start at $v^*$, we never move by more than $\sqrt{\alpha}$ from it. We now show that you can't even move $\sqrt{\alpha}$ without increasing the norm of $v$.

**Lemma E.2** (Restatement of Lemma 2.2). *Suppose $v_0 = v^*$. For any two time steps $0 \leq a < b \leq n$,*

$$\|P\widehat{v}_b - P\widehat{v}_a\|_2^2 \leq 50 \cdot \alpha \log(\|v_b\|_2 / \|v_a\|_2).$$

*Proof.* We have

$$\|P\widehat{v}_a\|_2 \leq \sqrt{\alpha} + \|Pv_0\|_2 / \|v_n\|_2$$
$$= \sqrt{\alpha} + \|Pv^*\|_2 / \|v_n\|_2$$
$$\leq \sqrt{\alpha}$$

where the first step follows from Lemma E.1, second step follows from $v_0 = v^*$ and the last step follows from definition of $P$ (see Definition C.2, which implies $Pv^* = 0$, see Claim C.3).

We can show

$$\|P\widehat{v}_b - P\widehat{v}_a\|_2^2 \leq (\|P\widehat{v}_b\|_2 + \|P\widehat{v}_a\|_2)^2$$
$$\leq (2\sqrt{\alpha})^2$$
$$\leq 4\alpha.$$

where the second step follows from $\|P\widehat{v}_b\|_2 \leq \sqrt{\alpha}$ and $\|P\widehat{v}_a\|_2 \leq \sqrt{\alpha}$.

Now, we can consider two cases.

**Case 1.** if $\log(\|v_b\|_2 / \|v_a\|_2) \geq 1$, then we already finished the proof.

**Case 2.** if $\log(\|v_b\|_2/\|v_a\|_2) < 1$. In the next paragraph, we will prove this case.

We define $w$ to be the unit vector in direction $P(\widehat{v}_b - \widehat{v}_a)$, i.e.,

$$w = P(\widehat{v}_b - \widehat{v}_a)/\|P(\widehat{v}_b - \widehat{v}_a)\|_2.$$

Using Lemma E.1, we can show the following thing,

$$
\begin{aligned}
&\langle v_b - v_a, w\rangle^2 \\
&= (\sum_{i=a+1}^{b} \eta\langle\phi(x_i), v_{i-1}\rangle\langle\phi(x_i), w\rangle)^2 \\
&\leq (\sum_{i=a+1}^{b} \eta\langle\phi(x_i), v_{i-1}\rangle^2)(\eta\sum_{i=a+1}^{b}\langle\phi(x_i), w\rangle^2) \\
&\leq (\sum_{i=a+1}^{b} \|v_i\|_2^2 \cdot \eta\langle\phi(x_i), \widehat{v}_{i-1}\rangle^2)(\eta\sum_{i=1}^{n}\langle\phi(x_i), w\rangle^2) \\
&\leq (\|v_b\|_2^2 \cdot \sum_{i=a+1}^{b} \eta\langle\phi(x_i), \widehat{v}_{i-1}\rangle^2)(\eta\sum_{i=1}^{n}\langle\phi(x_i), w\rangle^2) \\
&\leq (\|v_b\|_2^2 \cdot \sum_{i=a+1}^{b} \eta\langle\phi(x_i), \widehat{v}_{i-1}\rangle^2) \cdot \alpha \\
&\leq \|v_b\|_2^2 \cdot \log(\|v_b\|_2^2/\|v_a\|_2^2) \cdot \alpha.
\end{aligned}
\tag{10}
$$

where the first step follows from Property 5 of Claim D.1, the second step follows from Cauchy-Shwarz inequality, the third step follows from Definition C.5, the fourth step follows from $\|v_i\|_2 \leq \|v_b\|_2$ for all $i \leq b$ (see Property 2 of Claim D.1), the fifth step follows from the definition of $\alpha$, and the last step follows from $\log(\|v_b\|_2^2/\|v_a\|_2^2) \geq \sum_{i=a+1}^{b} \eta\langle x_i, \widehat{v}_{i-1}\rangle^2$ for all $a < b$ (see Property 4 of Claim D.1).

Therefore, we can upper bound $\|P\widehat{v}_b - P\widehat{v}_a\|_2^2$ in the following sense,

$$
\begin{aligned}
&\|P\widehat{v}_b - P\widehat{v}_a\|_2^2 \\
&= \langle\widehat{v}_b - \widehat{v}_a, w\rangle^2 \\
&= \langle\widehat{v}_b - \frac{\|v_a\|_2}{\|v_b\|_2}\widehat{v}_a + \frac{\|v_a\|_2}{\|v_b\|_2}\widehat{v}_a - \widehat{v}_a, w\rangle^2 \\
&\leq 2\langle\widehat{v}_b - \frac{\|v_a\|_2}{\|v_b\|_2}\widehat{v}_a, w\rangle^2 + 2\langle\frac{\|v_a\|_2}{\|v_b\|_2}\widehat{v}_a - \widehat{v}_a, w\rangle^2
\end{aligned}
\tag{11}
$$

where the first step follows from the definition of $w$, the second step follows from adding a term and minus the same term, and the last step follows from $\langle a+b, c\rangle^2 \leq 2\langle a, c\rangle^2 + 2\langle b, c\rangle^2$ (This is just triangle inequality and applying to each coordinate of the vector.).

For the first term in the above equation Eq. (11) (ignore the constant factor 2), we have

$$
\begin{aligned}
\langle\widehat{v}_b - \frac{\|v_a\|_2}{\|v_b\|_2}\widehat{v}_a, w\rangle^2 &= \langle\frac{v_b}{\|v_b\|_2} - \frac{\|v_a\|_2}{\|v_b\|_2}\widehat{v}_a, w\rangle^2 \\
&= \langle\frac{v_b}{\|v_b\|_2} - \frac{v_a}{\|v_b\|_2}, w\rangle^2 \\
&= \frac{1}{\|v_b\|_2^2} \cdot \langle v_b - v_a, w\rangle^2 \\
&\leq \alpha \cdot \log(\|v_b\|_2^2/\|v_a\|_2^2) \\
&= 2\alpha \cdot \log(\|v_b\|_2/\|v_a\|_2)
\end{aligned}
\tag{12}
$$

where the first step follows from definition of $\widehat{v}_b$, the second step follows from definition of $\widehat{v}_a$ (see Definition C.5), the fourth step follows from Eq. (10).

For the second term of that equation Eq. (11) (ignore the constant factor 2), we have

$$
\begin{aligned}
\langle \frac{\|v_a\|_2}{\|v_b\|_2}\widehat{v}_a - \widehat{v}_a, w \rangle^2 &= (\frac{\|v_a\|_2}{\|v_b\|_2} - 1)^2 \cdot \langle \widehat{v}_a, w \rangle^2 \\
&= (\frac{\|v_a\|_2}{\|v_b\|_2} - 1)^2 \cdot \langle \widehat{v}_a, P(\widehat{v}_b - \widehat{v}_a) \rangle^2 \\
&= (\frac{\|v_a\|_2}{\|v_b\|_2} - 1)^2 \cdot \langle P\widehat{v}_a, P(\widehat{v}_b - \widehat{v}_a) \rangle^2 \\
&\le (\frac{\|v_a\|_2}{\|v_b\|_2} - 1)^2 \cdot 4\|P\widehat{v}_a\|_2^2 \\
&\le (\frac{\|v_a\|_2}{\|v_b\|_2} - 1)^2 \cdot 4\alpha \\
&\le 4\log(\frac{\|v_b\|_2}{\|v_a\|_2}) \cdot 4\alpha
\end{aligned}
\tag{13}
$$

where the second step follows from definition of $w$, the third step follows from $P = P^2$ (then $\langle\langle a, P^2 b \rangle = a^\top PPb = \langle Pa, Pb \rangle)$, the fourth step follows from that both $\widehat{v}_a$ and $\widehat{v}_b$ are unit vectors, the fifth step follows from $\|P\widehat{v}_a\|_2 \le \sqrt{\alpha}$, the last step follows from $(\frac{1}{x} - 1)^2 \le 4\log x$ for all $x \in [1, 2]$ (Note that, here we treat $x = \|v_b\|_2/\|v_a\|_2$. The reason why we can assume $x \ge 1$ is due to Property 2 of Claim D.1. The reason why we can assume $x \le 2$ is due to this case we restrict $\log(x) \le 1$, which implies that $x \le 2$.).

Thus,

$$
\begin{aligned}
&\|P\widehat{v}_b - P\widehat{v}_a\|_2^2 \\
&\le 2\langle \widehat{v}_b - \frac{\|v_a\|_2}{\|v_b\|_2}\widehat{v}_a, w \rangle^2 + 2\langle \frac{\|v_a\|_2}{\|v_b\|_2}\widehat{v}_a - \widehat{v}_a, w \rangle^2 \\
&\le 2 \cdot 2\alpha \log(\|v_b\|_2/\|v_a\|_2) + 2 \cdot 16\alpha \log(\|v_b\|_2/\|v_a\|_2) \\
&\le 50\alpha \log(\|v_b\|_2/\|v_a\|_2).
\end{aligned}
$$

where the first step follows from Eq. (11), and the second step follows from Eq. (12), and Eq. (13).

Now, we complete the proof. $\qquad\square$

### E.3. Results on Sequences

**Claim E.3.** *Let $a \in \mathbb{R}^n$ and assume that $a_1 = 0$. For each $j \in [n]$ and $k \in [\log n]$, we define*

$$b_{j,k} := a_{1 + 2^k \cdot j}$$

*Note that, if $1 + 2^k \cdot j > n$, then we assume that $b_{j,k} = 0$.*

*Then, we have*

$$
\max_{j \in [n]} a_j^2 \le (\log n) \sum_{k=0}^{(\log n)-1} \sum_{j=1}^{n} (b_{j,k} - b_{j-1,k})^2.
$$

*Proof.* We define $j^* := \arg\max_{j \in [n]} a_j^2$.

We define $j_k := 1 + 2^k \lfloor \frac{j^*-1}{2^k} \rfloor$.

According to the definition of $j_k$, we have that

$$
j_0 = 1 + 2^0 \lfloor \frac{j^* - 1}{2^0} \rfloor = j^*
$$

and

$$j_{\log n} = 1 + 2^{\log n} \lfloor \frac{j^* - 1}{2^{\log n}} \rfloor = 1.$$

Thus,

$$
\begin{aligned}
a_{j^*} &= a_{j^*} - a_1 \\
&= a_{j_0} - a_{j_{\log n}} \\
&= \sum_{k=0}^{(\log n)-1} (a_{j_k} - a_{j_{k+1}})
\end{aligned}
\tag{14}
$$

where the first step follows from the definition of $a_1 = 0$.

Let $j_k = 1 + 2^k y$ and $j_{k+1} = 1 + 2^{k+1} z$. It is obvious that $2z \geq y \geq z$. Using Fact B.3, we know that $|2z - y| \leq 1$.

Now, we consider two cases.

**Case 1.** $j_k = j_{k+1}$. In this case, we have

$$a_{j_k} - a_{j_{k+1}} = 0.$$

**Case 2.** $j_k \neq j_{k+1}$.

Then we have

$$
\begin{aligned}
a_{j_k} - a_{j_{k+1}} &= b_{y,k} - b_{2z,k} \\
&= (b_{y,k} - b_{y+1,k}).
\end{aligned}
$$

Thus,

$$
\begin{aligned}
a_{j^*}^2 &= ( \sum_{k=0}^{(\log n)-1} (a_{j_k} - a_{j_{k+1}}) )^2 \\
&\leq (\log n) \cdot \sum_{k=0}^{(\log n)-1} (a_{j_k} - a_{j_{k+1}})^2 \\
&\leq (\log n) \cdot \sum_{k=0}^{(\log n)-1} \cdot \sum_{j=1}^{n} (b_{j,k} - b_{j-1,k})^2
\end{aligned}
$$

where the first step follows from Eq. (14), and the second step follows from our definition of $j_k$ for proof. $\square$

**Lemma E.4.** *Let $A \in \mathbb{R}^{d \times n}$ have first column all zero, i.e., for all $i \in [d]$, $A_{i,1} = 0$. For each $j \in [n]$ and $k \in [\log n]$, define $b_{j,k}$ to be column $1 + 2^k \cdot j$ of $A$. If $1 + 2^k \cdot j > n$, then we assume $b_{j,k}$ is a zero column.*

- *Property 1. For each $i \in [d]$, we have*

$$\max_{j \in [n]} A_{i,j}^2 \leq (\log n) \sum_{k=0}^{\log n} \sum_{j=2}^{n+1} (b_{j,k} - b_{j-1,k})_i^2$$

- *Property 2. Then:*

$$\sum_{i=1}^{d} \max_{j \in [n]} A_{i,j}^2 \leq (\log n) \sum_{k=0}^{\log n} \sum_{j=2}^{n+1} \|b_{j,k} - b_{j-1,k}\|_2^2$$

*Proof.* Using Claim E.3, we can prove Property 1.

Applying Claim E.3 for $d$ different rows, we have

$$\sum_{i=1}^{d} \max_{j \in [n]} A_{i,j}^2 \le (\log n) \sum_{k=0}^{\log n} \sum_{j=2}^{n+1} \|b_{j,k} - b_{j-1,k}\|_2^2.$$

Thus, we have proved property 2.

$\square$

## E.4. Upper Bound for the Summation of Inner Product

We return to the streaming PCA setting. The goal of this section is to show that, if $v_0 = v^*$, then $\|v_n\|_2$ is large.

**Lemma E.5** (Restatement of Lemma 2.3). *If $v_0 = v^*$, then for $i \in [n]$, we have*

$$\eta \sum_{i=1}^{n} \langle \phi(x_i), P\widehat{v}_{i-1} \rangle^2 \le 100 \cdot \alpha^2 \cdot \log^2 n \cdot \log \|v_n\|_2.$$

*Proof.* For $i \in [n]$, we define $u_i := P\widehat{v}_i$. This also means $\|u_i\|_2 \le 1$.

Since $u_i$ lies in span of $P$ and by Claim C.3 that $Pv^* = 0$, we know that $u_i \perp v^*$.

Hence, we have

$$\langle u_i, v^* \rangle = 0.$$

For each $i \in [d]$, for each $j \in [n]$, we define a matrix $A_{i,j} \in \mathbb{R}^{d \times n}$ as follows

$$A_{i,j} := \langle \phi(x_i), u_{j-1} \rangle.$$

We can show

$$\sum_{i=1}^{d} \max_{j \in [n]} \langle \phi(x_i), u_j \rangle^2$$

$$\le (\log n) \sum_{k=0}^{\log n} \sum_{j=2}^{n+1} \|b_{j,k} - b_{j-1,k}\|_2^2$$

$$= (\log n) \sum_{k=0}^{\log n} \sum_{j=2}^{n+1} ((b_{j,k})_i - (b_{j-1,k})_i)^2$$

$$= (\log n) \sum_{k=0}^{\log n} \sum_{j=2}^{n+1} \sum_{i=1}^{d} (\langle \phi(x_i), u_{2^k j} \rangle - \langle \phi(x_i), u_{2^k(j-1)} \rangle)^2. \tag{15}$$

where the first step follows from Lemma E.4, the second step follows from definition of $\ell_2$ norm, the third step follows from $(b_{j,k})_i = A_{i,1+2^k \cdot j} = \langle \phi(x_i), u_{1+2^k \cdot j-1} \rangle = \langle \phi(x_i) u_{2^k \cdot j} \rangle$.

For each $(k, j)$-term in the above equation, we have

$$\sum_{i=1}^{d} (\langle \phi(x_i), u_{2^k j} \rangle - \langle \phi(x_i), u_{2^k(j-1)} \rangle)^2$$

$$= \sum_{i=1}^{d} (\langle \phi(x_i), u_{2^k j} - u_{2^k(j-1)} \rangle)^2$$

$$\le \frac{\alpha}{\eta} \cdot \|u_{2^k j} - u_{2^k(j-1)}\|_2^2. \tag{16}$$

where the first step follows from simple algebra, the second step follows from $\langle u_i, v^* \rangle = 0$ and $\|u_i\|_2$ for all $i \in [n]$ and Property 3 of Definition C.1.

Then, for each $k \in [\log n]$, we have

$$\sum_{j=2}^{n+1} \|u_{2^k j} - u_{2^k(j-1)}\|_2^2 \leq 50\alpha \log \frac{\|v_n\|_2}{\|v_0\|_2}$$

$$= 50\alpha \log \|v_n\|_2 \tag{17}$$

where the first step follows from summation over $j \in [2, n+1]$ by Lemma E.2 for each $j$, and the second step follows from $v_0 = v^*$ (see assumption in statement of Lemma E.5) and $\|v^*\|_2 = 1$.

Thus,

$$\eta \sum_{i=1}^{d} \langle \phi(x_i), P\widehat{v}_{i-1} \rangle^2$$

$$\leq \eta \sum_{i=1}^{d} \max_{j \in [n]} \langle \phi(x_i), u_j \rangle^2$$

$$= \eta(\log n) \sum_{k=0}^{\log n} \sum_{j=2}^{n+1} \sum_{i=1}^{d} (\langle \phi(x_i), u_{2^k j} \rangle - \langle \phi(x_i), u_{2^k(j-1)} \rangle)^2$$

$$= \alpha(\log n) \sum_{k=0}^{\log n} \sum_{j=2}^{n+1} \|u_{2^k j} - u_{2^k(j-1)} \rangle\|_2^2$$

$$\leq (\log n) \sum_{k=0}^{\log n} 50\alpha^2 \log \|v_n\|_2$$

$$\leq 100 \cdot \alpha^2 \cdot \log^2 n \cdot \log \|v_n\|_2$$

where the first step follows from our definition for this proof, the second step follows from Eq. (15), the third step follows from Eq. (16), the fourth step follows from Eq. (17), and the last step follows from simple algebra.

Therefore, we complete the proof. $\qquad\square$

### E.5. Lower bound on Log of Norm

**Lemma E.6** (Restatement of Lemma 2.4)**.** *Let $\alpha \in (0, 0.1)$. Let $C_1 \geq 200$ denote some fixed constant. Then if $v_0 = v^*$ we have*

$$\log(\|v_n\|_2) \geq \frac{\beta/8}{1 + C_1 \cdot \alpha^2 \log^2 n}.$$

*Further, if $\alpha \in (0, 1/(10 C_1 \log n))$, we have*

$$\|v_n\|_2 \geq \exp(\beta/20).$$

*Proof.* We rewrite $\widehat{v}_i = a_i \cdot v^* + u_i$ for $u_i \perp v^*$.

Then, we have

$$\langle \phi(x_i), \widehat{v}_{i-1} \rangle^2$$

$$= \langle \phi(x_i), a_{i-1} \cdot v^* + u_{i-1} \rangle^2$$

$$\geq \frac{a_{i-1}^2}{2} \langle \phi(x_i), v^* \rangle^2 - \langle \phi(x_i), u_{i-1} \rangle^2. \tag{18}$$

where the second step follows from Claim B.5.

Applying Lemma E.1 with $v_0 = v^*$, we have

$$\|P\widehat{v}_i\|_2^2 \leq \alpha. \tag{19}$$

Note that

$$
\begin{aligned}
\|P\widehat{v}_i\|_2^2 &= \|P(a_i v^* + u_i)\|_2^2 \\
&= \|Pu_i\|_2^2 \\
&= \|u_i\|_2^2 \\
&\geq \frac{1}{2}\|\widehat{v}_i\|_2^2 - \|a_i v^*\|_2^2 \\
&= \frac{1}{2} - a_i^2
\end{aligned}
\tag{20}
$$

where the first step follows from our definition of $\widehat{v}_i = a_i \cdot v^* + u_i$, the second step follows from $Pv^* = 0$ (see Claim C.3), the third step follows from the definition of $P$, the fourth step follows from Claim B.5, and the last step follows from simple algebra.

Thus, we have

$$
\begin{aligned}
a_i &\geq (\frac{1}{2} - \alpha)^{1/2} \\
&\geq \frac{1}{2} - \alpha
\end{aligned}
\tag{21}
$$

where the first step follows from combining Eq. (19) and Eq. (20), and the last step follows from $\alpha \in (0, 0.1)$.

Now, summing up over $i \in [n]$, we get

$$
\begin{aligned}
&\eta \sum_{i=1}^n \langle \phi(x_i), \widehat{v}_{i-1} \rangle^2 \\
&\geq \eta \sum_{i=1}^n (\frac{a_{i-1}^2}{2} \langle \phi(x_i), v^* \rangle^2 - \langle \phi(x_i), u_{i-1} \rangle^2) \\
&\geq \frac{1}{4}\beta - \eta \sum_{i=1}^n \langle \phi(x_i), u_{i-1} \rangle^2.
\end{aligned}
$$

where the first step follows summing over $i \in [n]$ from Eq. (18) for each $i$, and the second step follows from Eq. (21).

We can lower bound $\log(\|v_n\|_2)$ as follows:

$$
\begin{aligned}
\log\|v_n\|_2 &\geq \frac{1}{2}\eta \sum_{i=1}^n \langle \phi(x_i), \widehat{v}_{i-1} \rangle^2 \\
&\geq \frac{1}{8}\beta - C_1 \cdot \alpha^2 \log^2 n \log\|v_n\|_2,
\end{aligned}
$$

where the first step follows from Lemma E.7, the second step follows from Lemma E.5 with $C_1 \geq 200$ is a sufficiently large constant.

The above equation implies the following

$$\log\|v_n\|_2 \geq \frac{\beta/8}{1 + C_1 \cdot \alpha^2 \log^2 n}.$$

$\square$

## E.6. Lower Bound of $\|v_n\|_2$

**Lemma E.7** (Restatement of Lemma 2.5). *We have*

$$\|v_n\|_2 \geq \sqrt{\eta} \cdot (\sum_{i=1}^{n} \langle \phi(x_i), v_{i-1} \rangle^2)^{1/2}$$

*Proof.* We define

$$B_i := \|v_i\|_2^2,$$

We also define

$$A_i := \log \frac{B_i}{B_{i-1}}$$

Then using Property 3 of Claim D.1, it is easy to see that

$$A_i \geq \eta \langle \phi(x_i), \widehat{v}_{i-1} \rangle^2.$$

Thus,

$$
\begin{aligned}
A_i \cdot B_{i-1} &\geq \eta \langle \phi(x_i), \widehat{v}_{i-1} \rangle^2 \cdot B_{i-1} \\
&\geq \eta \langle \phi(x_i), \widehat{v}_{i-1} \rangle^2 \cdot \|v_{i-1}\|_2^2 \\
&= \eta \langle \phi(x_i), v_{i-1} \rangle^2
\end{aligned}
$$

where the third step follows from Definition C.5.

Therefore, we can show the following things,

$$
\begin{aligned}
\eta \sum_{i=1}^{n} \langle \phi(x_i), v_{i-1} \rangle^2 &\leq \sum_{i=1}^{n} A_i B_{i-1} \\
&\leq B_n \\
&= \|v_n\|_2^2
\end{aligned}
$$

where the first step follows from $\eta \langle \phi(x_i), v_{i-1} \rangle^2 \leq A_i B_{i-1}$, the second step follows from Claim B.4, and the third step follows from our definition for proof. $\qquad \square$

