# OpenReview forum: "Streaming Kernel PCA Algorithm With Small Space"
_CPAL.cc/2025/Proceedings_Track — CPAL 2025 (Proceedings Track) Poster_

### Official Review · Reviewer_M9Hy · 2025-01-06
**This paper introduces a streaming kernel PCA algorithm that operates within small memory spaces, making it suitable for large-scale data processing. The proposed method extends Oja's algorithm to kernel PCA, utilizing an iterative update rule that efficiently handles streaming data by maintaining the principal components dynamically.**

**Rating:** 7
**Confidence:** 2

**Review:**

1. This paper successfully extends traditional PCA to kernel PCA in a streaming context, which is a big contribution given the computational constraints of large data sets.

2.The paper provides a solid theoretical foundation, including a detailed analysis of the convergence properties of the algorithm, which is crucial for establishing its reliability.

While the theoretical analysis is robust, the paper lacks extensive empirical results to demonstrate the practical effectiveness and efficiency of the proposed algorithm under various real-world conditions.

---

### Official Review · Reviewer_XLUm · 2025-01-11
**New kernel PCA algorithm with low space need**

**Rating:** 7
**Confidence:** 3

**Review:**

Streaming method on traditional PCA reduces computational cost and memory usage. This paper provides an streaming method for kernel PCA. The proposed method has O(m+d), m is the dimension of kernel space and d is the dimension of input x. Space required for this algorithm has advantage in some scenarios comparing to some other streaming kernel PCA method. This paper also provides mathematical justification for the performance of the algorithm. It would be even better if some simulation results can be provided as well.

---

### Official Review · Reviewer_f2kY · 2025-01-24
**Clear Analysis and Rigorous Foundations: A Useful Approach to Streaming Kernel PCA**

**Rating:** 7
**Confidence:** 4

**Review:**

Strength:\\
1. This paper proposes a streaming kernel PCA algorithm that uses \( O(\cdot) \) space complexity. It incorporates Oja's streaming PCA method. The algorithm's performance is analyzed, demonstrating its ability to approximate the top principal component with low error under certain spectral ratio conditions.\\
2. The proofs in the paper are detailed and rigorous. The appendix sections (e.g., A, B, C, D) are comprehensive and include numerous useful definitions and claims.

Weakness:\\
1. Typo: In line 239, since \( v_0 \) is a vector, \( \mathcal{N}(0,1) \) should be replaced with \( \mathcal{N}(0, \bm{I}_d) \), or \( v_0 \) should be replaced with \( (v_0)_i \). In line 162, the inner product of two vectors should be squared. In line 574, \( v_0 = v^* \) would be clearer if equality were used rather than inequality in the second step.\\
2. The algorithm's performance relies heavily on the spectral ratio conditions (e.g., \( R \geq C \cdot \log n \cdot \log d \) in Theorem 3.2). This assumption may not hold for all real-world datasets, potentially limiting its applicability.

Question:\\
1. Is the notation "log" based on base 2? If so, in line 535, the application of Property 3 might not be trivial.\\

---

### Meta-Review · Area_Chair_1i3c · 2025-02-05

**Recommendation:** Accept (Poster)
**Confidence:** 5

**Metareview:**

This paper presents a streaming kernel PCA algorithm with efficient space complexity, leveraging Oja’s method for principal component approximation. The theoretical analysis is rigorous, supported by detailed proofs and a comprehensive appendix. While all reviewers recognize the contribution and its strong theoretical foundation, concerns are raised regarding the reliance on spectral ratio conditions and the limited empirical evaluation. Please address these comments and incorporate the necessary revisions into the final draft.

**Given the solid theoretical contributions and the positive comments from reviewers, I recommend acceptance.**

---

### Decision · Program_Chairs · 2025-02-11

Accept (Poster)